# Prevalence and epidemiological characteristics of COVID-19 after one year of pandemic in Jakarta and neighbouring areas, Indonesia: A single center study

**Wuryantari Setiadi**[1], **Ismail Ekoprayitno Rozi**[1], **Dodi Safari**[1], **Wa Ode Dwi Daningrat**[1], **Edison Johar**[1], **Benediktus Yohan**[1], **Frilasita Aisyah Yudhaputri**[1], **Karina Dian Lestari**[1], **Sukma Oktavianthi**[1], **Khin Saw Aye Myint**[1], **Safarina G. Malik**[1], **Amin Soebandrio**[1,2]\*, on behalf of the Wascove team[¶]

1 Eijkman Institute for Molecular Biology, National Research and Innovation Agency, Jakarta, Indonesia,
2 Faculty of Medicine, University of Indonesia, Jakarta, Indonesia

¶ Membership of the Wascove team is provided in the Acknowledgments
\* aminsoebandrio@eijkman.go.id

**Data Availability Statement:** All relevant data are within the manuscript.

## Abstract

We determined the prevalence and epidemiological characteristics of COVID-19 in Jakarta and neighboring areas, Indonesia from March 2020 to February 2021, based on nasopharyngeal/oropharyngeal (NP/OP) swab specimens that were tested at the Eijkman Institute for Molecular Biology, Jakarta. NP/OP swab specimens were collected from COVID-19 suspects or individuals in contact tracing programs from primary healthcare centers (PHC) and hospitals. The specimens were screened for the SARS-CoV-2 by qRT-PCR. Demography data and clinical symptoms were collected using national standardized laboratory form. Of 64,364 specimens, 10,130 (15.7%) were confirmed positive for SARS-CoV-2, with the peak prevalence of infection in March 2020 (26.3%) follow by in January 2021 (23.9%) and February 2021 (21.8%). We found that the positivity rate of the specimens from Jakarta, West Java, and Banten was 16.3%, 13.3%, and 16.8%, respectively. Positivity rate was higher in specimens from hospitals (16.9%) than PHC (9.4%). Of the positive specimens, 29.6% were from individuals aged >60 years old, followed by individuals aged 41–60 years old (24.2%). Among symptomatic cases of SARS-CoV-2, the most common symptoms were cough, fever, and a combination of both cough & fever. In conclusion, this study illustrates the prevalence and epidemiological characteristics from one COVID-19 diagnostic center in Jakarta and neighbouring areas in Indonesia.

## Introduction

SARS-CoV-2 infection was first reported in 2019, and has since spread throughout the world causing more than 208,470,375 COVID-19 cases and 4,377,979 deaths globally, as of August 18, 2021 [1]. Indonesia confirmed the first positive COVID-19 case on March 2, 2020 [2].

**Funding:** This work was supported by The Ministry of Research and Technology/National Research and Innovation Agency, Republic of Indonesia. The PCR reagents to test outbreak specimens were provided by U.S. Centers for Disease Control and Prevention (US CDC), Indonesian National Board for Disaster Management (Badan Nasional Penanggulangan Bencana; BNPB), Embassy of New Zealand in Indonesia, and Indonesian States Intelligence Agency (Badan Intelijen Negara Republik Indonesia; BIN). The funders had no role in study design, data collection and analysis, decision to publish, or preparation of the manuscript.

**Competing interests:** The authors have declared that no competing interests exist

Within 40 days, COVID-19 cases were reported by all provinces [3]. As of August, 2021, Indonesia had 3,892,479 confirmed COVID-19 cases, with 120,013 deaths [4].

In the early phase of COVID-19 pandemic, besides the vast territory, Indonesia was also challenged with limited laboratory infrastructure and capacity for COVID-19 diagnosis. Lack of test reagents, consumables, personal protective equipment, and shortage of human resources further exacerbated the situation [5–7]. The recent emergence of more transmissible variants, such as the Alpha and Delta variants, put an additional burden on the government effort to contain COVID-19 [8,9].

Eijkman Institute for Molecular Biology (EIMB) in Jakarta, Indonesia, was appointed by the Indonesian Ministry of Health as one of the regional laboratories to test for SARS-CoV-2 [5]. Since then, over 69,000 specimens have been submitted from primary healthcare centers (PHC) and hospitals mainly from Jakarta and neighboring areas. In this study, we reported the prevalence and epidemiological characteristics of COVID-19 cases from samples tested in EIMB from March 2020 to February 2021.

These findings could contribute to the diagnostic, preventive and curative measures of COVID-19 pandemic in Indonesia.

## Materials and method

### Study design and data collection

This was a retrospective study using of SARS-CoV-2 surveillance data conducted at EIMB, Jakarta as one of the referral laboratories for SARS-CoV-2 testing in Indonesia. This study was performed in accordance with the human subject protection guidance provided by the Eijkman Institute Research Ethics Committee No. 127. All data were fully anonymized before accessing the database. Nasopharyngeal/oropharyngeal (NP/OP) swabs, sputum, and serum specimens were submitted by PHCs and hospitals in Jakarta Province (n = 25,314) and neighbouring areas in Indonesia including Banten Province (Tangerang [n = 11,901], South Tangerang [n = 4,519], and Serang [n = 6], West Java Province (Bekasi [n = 11,385], Bogor [n = 5,577], Depok [n = 3,865], Purwakarta [n = 162], Karawang [n = 114], Bandung [n = 4]). We also received a number of specimens from other provinces such as North Sulawesi (n = 1,324), South Sulawesi (n = 11), Riau (n = 3), Papua (n = 5), East Nusa Tenggara (n = 82), Lampung (n = 261), East Kalimantan (n = 1), Central Kalimantan (n = 50), Central Java (n = 8), Jambi (n = 18), Yogyakarta (n = 6), and Bali (n = 1).

All NP/OP swab specimens were tested for SARS-CoV-2 by quantitative reverse-transcription-polymerase chain reaction (qRT-PCR) assay based on Charité Institute of Virology Universitätsmedizin Berlin (modified), US-Centers for Disease Control (modified), and Hong Kong University, in accordance with WHO, or with Cobas® 6800 automated nucleic acid test system (Roche USA) [10–12]. Demographic and clinical data were collected using national standard laboratory form that include symptom onset date, specimen collection date, contact history in the past 14 days, clinical manifestations, underlying health conditions, contact tracing, and community screening. Symptoms or clinical manifestation were defined based on self-reported symptoms by subjects except for pneumonia. Chest X-Ray results were used to defined pneumonia [13,14]

### Data analysis

Statistical analyses were carried out in Stata Software. Univariate analysis using Pearson's Chi-Square or Fisher's exact test was performed to determined factors associated with the COVID-19 positive cases. Association between symptoms with age and gender were performed using Pearson's Chi-Square. Stratification analysis was performed to compare symptoms between

samples acquired from hospitals and PHCs. Multivariate analysis was performed to identify factors associated with the COVID-19 positive cases, and controlled in fitted model using multiple logistic regression.

## Results

From March 12, 2020, to February 26, 2021, a total of 64,364 NP/OP swab specimens were tested at the EIMB for SARS-CoV-2 nucleic acid test from health facilities in Jakarta, neighbouring areas and other provinces (Fig 1). The majority of the specimens were from three provinces: Jakarta (n = 25,174; 39.1%), West Java (n = 20,993; 32.6%), and Banten (n = 16,394; 25.5%) (Fig 1). Most of the NP/OP swab specimens were collected from hospital (n = 54,372; 84.5%) and followed by PHC (n = 8,701; 13.5%). NP/OP swabs from Provincial and District Health Offices, Private Clinics and Laboratories were accounted for 2.0% of all specimens (n = 1,258) (Table 1). We identified, 95.5% of specimens were from adults aged >18 years old (n = 61,403), and 61.6% (n = 39,477) were from females.

Of the 64,364 swab specimens tested, 15.7% (n = 10,130) were positive for SARS-CoV-2. The percentage of COVID-19 positive cases varied from March 2020 to February 2021 (Fig 2). The highest positive rate were observed in the first month of the pandemic (March 2020; 652/ 2,481; 26.3%) followed by early 2021 (January 2021; 1,543/6,415; 24.1%, and February 2021;

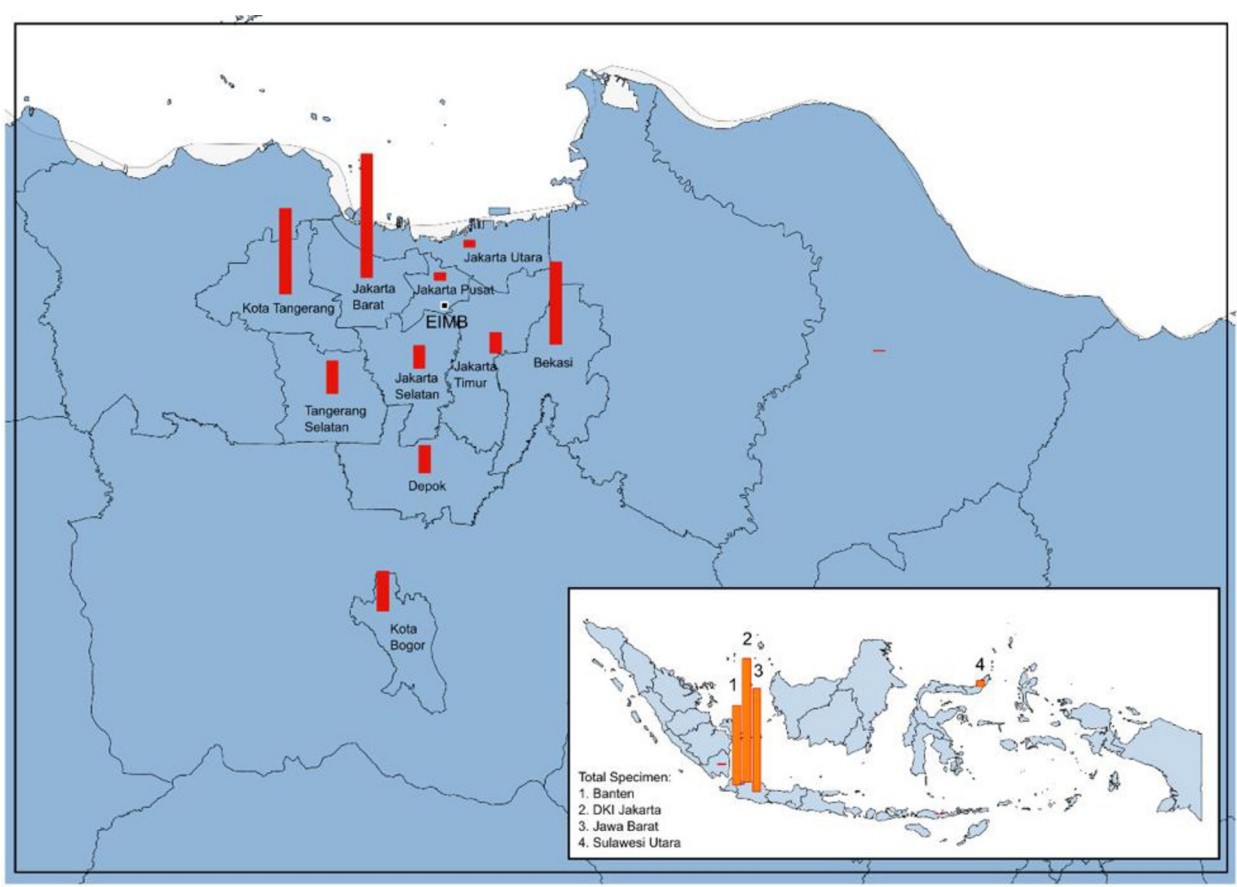

**Fig 1. NP/OP swab specimens were submitted to EIMB form health facilities in different regions of Indonesia from March 2020 –February 2021.** The majority of the specimens were from three provinces: Jakarta (n = 25,314; 39.2%) followed by West Java especially Bogor, Bekasi, and Depok (n = 21,107; 32.6%), and Banten especially South Tangerang and Tangerang (n = 16,426; 25.4%). (https://www.naturalearthdata.com/).

**Table 1. Demographics and characteristics of COVID-19 suspects, or individuals in contact tracing program with eligible specimens collected at EIMB, from March 2020 to February 2021.**

| Variables | Category | SARS-CoV-2 | | | crude OR (95% CI) | p-Value | Adjusted OR (95% CI) | p-Value |
|---|---|---|---|---|---|---|---|---|
| | | Positive | Negative | % Positive | | | | |
| **Age** | | | | | | | | |
| | **0–5** | 126 | 1,021 | 11.0% | ref | ref | ref | ref |
| | **6–18** | 271 | 1,445 | 15.8% | 1.5 (1.2–1.9) | <0.001 | 2.1 (1.0–4.3) | 0.053 |
| | **19–40** | 4,676 | 36,751 | 11.3% | 1.0 (0.9–1.2 | 0.750 | 1.6 (0.8–3.1) | 0.215 |
| | **41–60** | 3,899 | 12,214 | 24.2% | 2.6 (2.1–3.1) | <0.001 | 3.3 (1.7–6.7) | 0.001 |
| | **>60** | 1,143 | 2,720 | 29.6% | 3.4 (2.8–4.1) | <0.001 | 3.7 (1.8–7.5) | <0.001 |
| **Gender** | | | | | | | | |
| | **Female** | 5,527 | 33,950 | 14.0% | ref | ref | ref | ref |
| | **Male** | 4,549 | 20,068 | 18.5% | 1.4 (1.3–1.5) | <0.001 | 1.2 (1.1–1.2) | <0.001 |
| **Occupation** | | | | | | | | |
| | **Healthcare Workers** | 1,348 | 17,095 | 7.3% | ref | ref | ref | ref |
| | **Non-Healthcare Worker** | 3,508 | 16,219 | 17.8% | 2.7 (2.6–2.9) | <0.001 | 2.6 (2.4–2.8) | <0.001 |
| **Type of Facilities** | | | | | | | | |
| | **PHC** | 814 | 7,887 | 9.4% | ref | ref | ref | ref |
| | **Hospital** | 9,176 | 45,196 | 16.9% | 2.0 (1.8–2.1) | <0.001 | 4.1 (3.6–4.7) | <0.001 |
| | **Other\*** | 138 | 1,120 | 11.0% | 1.2 (1.0–1.4) | 0.069 | 1.5 (1.0–2.1) | 0.043 |
| **Province** | | | | | | | | |
| | **Jakarta** | 4,111 | 21,063 | 16.3% | 3.0 (2.0–4.5) | <0.001 | ref | ref |
| | **Banten** | 2,182 | 14,212 | 13.3% | 2.4 (1.6–3.5) | <0.001 | 2.9 (1.8–4.7) | <0.001 |
| | **West Java** | 3,518 | 17,475 | 16.8% | 3.1 (2.1–4.6) | <0.001 | 2.3 (1.4–3.8) | 0.001 |
| | **North Sulawesi** | 290 | 1,034 | 21.9% | 4.4 (2.9–6.6) | <0.001 | 2.7 (1.7–4.4) | <0.001 |
| | **Other\#** | 27 | 419 | 6.1% | ref | ref | 1.9 (0.2–16.6) | 0.543 |

\*Provincial & District Health Offices, Clinic Laboratorium.

\#Lampung, Bali, DI Yogyakarta, Jambi, Central Java, Central Kalimantan, East Kalimantan, East Nusa Tenggara, Papua, Riau, South Sulawesi and North Sumatera.

1,039/4,757; 21.8%). The lowest percentage of positive case was observed in May 2020 (134/2,952; 4.5%) followed by the month of June 2020 (522/8,773; 6.0%), and July 2020 (801/7,798; 10,3%) (Fig 2).

The positive rate for COVID-19 was higher in >60 years of age group (1,143/3,863; 29.6%) followed by 41–60 years of age group (3,899/16,113; 24.2%) (Table 1). The lowest COVID-19 positive rate was observed in children under five (126/1,147; 11.0%) as compared to other age groups (Table 1). the specimens collected from hospitals had higher COVID-19 positivity rate (9,176/54,372; 16.9%) compared to PHC (814/8,701; 9.4%). (Table 1).

Chest X-ray data were available from 845 out of 10130 positive cases, which were reported as: pneumonia 679/845 (80.4%), other abnormalities (including bronchitis, pleural effusion, lung abscess, and increased bronchovascular markings) 46/845 (5.4%), and normal 120/845 (14.2%). Most of SARS-CoV-2 cases were asymptomatic (74.0%). Among symptomatic cases of SARS-CoV-2, we found that the most common reported clinical symptoms (or combination of symptoms) (n = 1,577) were cough (20.9%), a combination of both cough & fever (7.0%), and fever (6.8%) (Fig 3).

Subjects aged above 60 years old had a 3.4-fold higher risk of COVID-19 compare to other age groups (95% confidence interval (CI): 2.8–4.1). This followed by subjects aged 41–60 years old with odds ratio (OR) = 2.6 (95%CI: 2.1–3.1). In a fitted model, these results were consistently increased with ORs of 3.7 (95%CI 1.8–7.5) and 3.3 (95%CI 1.7–6.7), respectively. In this

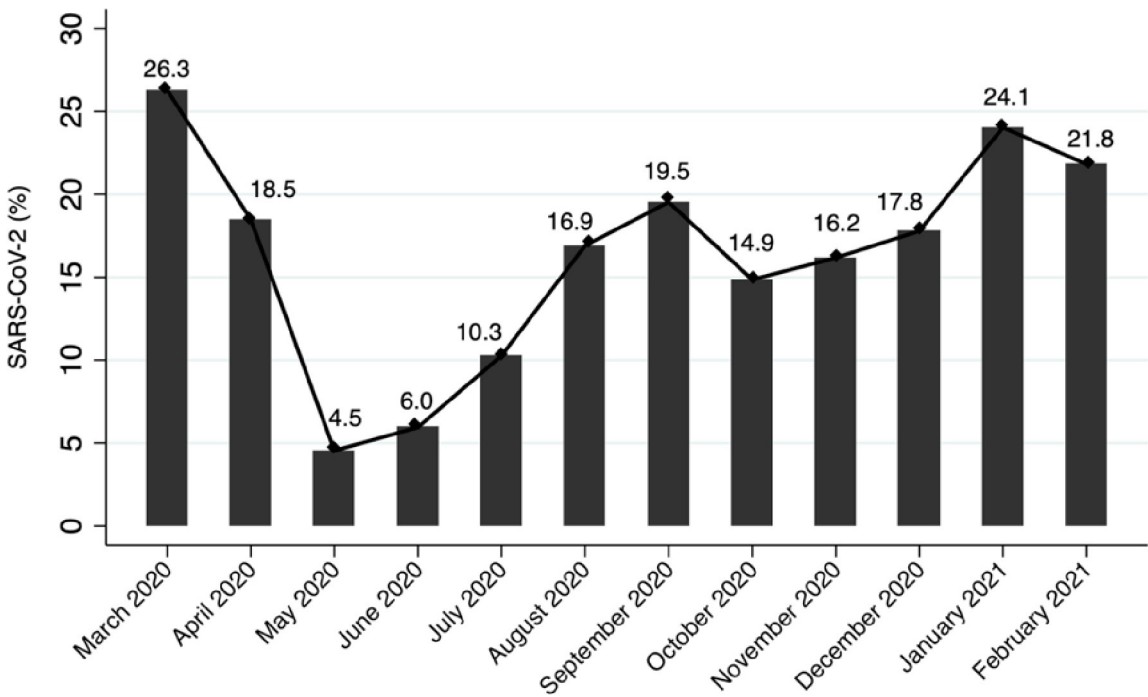

**Fig 2. SARS-CoV-2 positivity rate during the first year of pandemic in Jakarta, neighbouring areas, and other provinces in Indonesia from March 2020 –February 2021.**

analysis, non-healthcare workers showed a 2.7-fold increase in the risk to COVID-19 than healthcare workers (95% CI: 2.6–2.9). When controlled with other factors in a fitted model, the odds were slightly lower with OR 2.6 (95% CI: 2.4–2.8). Hospital-based specimens have higher positive results of SARS-CoV-2 compare to those from PHC or private clinics or laboratories with OR 2.0 (95%CI: 1.8–2.1). The odds were doubled when other factors were considered with OR 4.1 (95%CI: 3.6–4.7).

## Discussion

In this study, we described prevalence and epidemiological characteristic of COVID-19 in Jakarta, and its surrounding areas after the COVID-19 first case was confirmed in Indonesia in March 2020. As one of the appointed laboratories for COVID-19 detection, we tested more than sixty thousand NP/OP swab specimens from COVID-19 suspects or individuals under contact tracing program within the first year of the pandemic. A proportion of 15.7% of those specimens were positive for the COVID-19 positivity rates range from 4.5% (May 2020) to 26.5% (March 2020). As per 22 August 2021, The National Committee for Corona Virus Disease 2019 Handling and Economic Recovery (*Komite Penanganan Covid-19 dan Pemulihan Ekonomi Nasional*) reported that the five provinces with the highest COVID-19 confirmed cases (as % of the total national cases) are: Jakarta (21.3%), West Java (16.8%), Central Java (11.6%), East Java (9.4%), and East Kalimantan (3.7%) [15]. These provinces contributed to over 60% of the national cases six months after the COVID-19 pandemic [3,15].

In this study, we found that the lowest positive rate of COVID-19 was in May 2020 (4.5%). We suggest this condition was due to strict implementation of the public health social measures including the mask wearing, mobility restriction, and school or business closure [4]. In addition, the observance of Ramadan session which started on 23 April 2020 to 23 May 2020 is

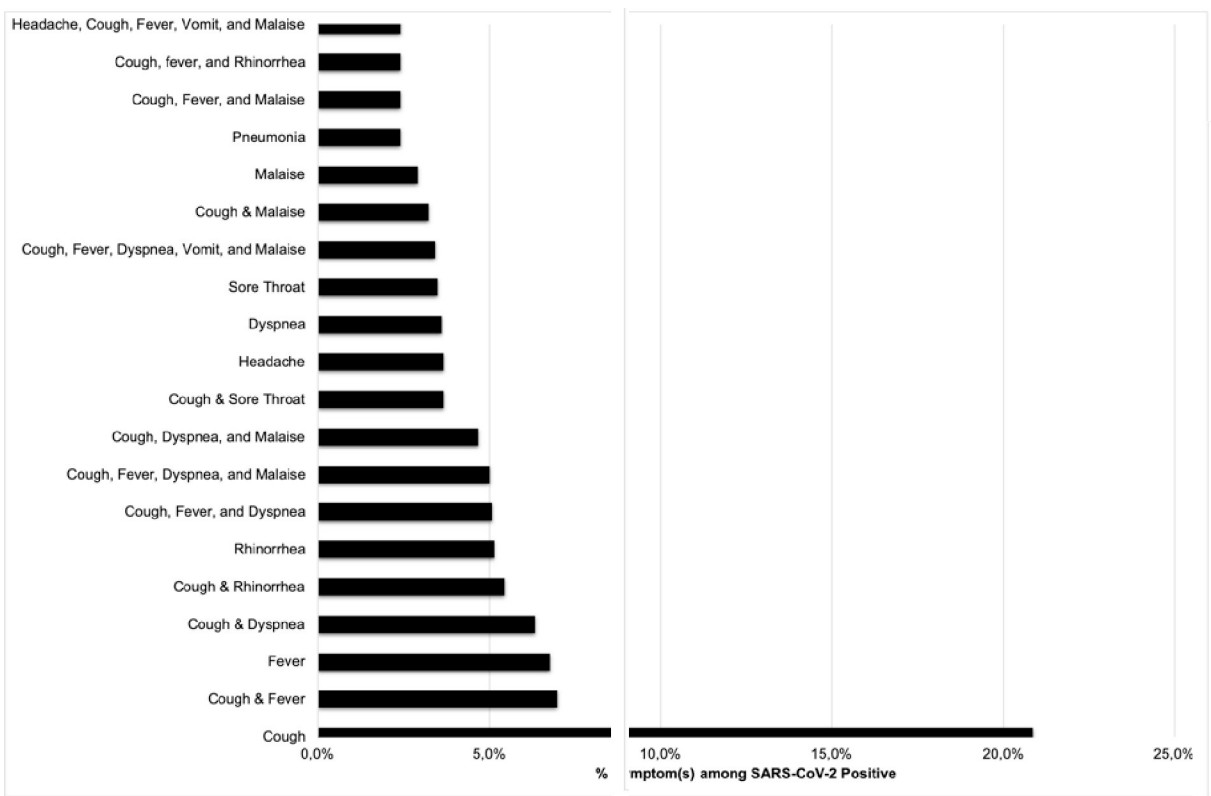

**Fig 3. Most common symptoms among PCR confirmed SARS-CoV-2 subjects in Indonesia from March 2020 –February 2021.** Chest X-Ray results were used to defined pneumonia.

also possible to contribute to the decrease of the positivity rate. After Ramadan, however, the COVID-19 positivity rate started to raise again from 6% in June to 17% in December 2020. Although the Government has made efforts to limit holidays and travel to other cities during the holiday season and New Year's celebrations, the positive rate of COVID-19 increased in the month of January and February 2021 to 23.9% and 21.4%, respectively. As reported previously, COVID-19 cases started to increase one or two weeks following holiday sessions [16].

In this study, we also found that the subjects aged over 40 years old had a higher positive rate of COVID-19 than other age groups, while a lower positive rate of COVID-19 were observed in children under the age of five. This finding is in line with other studies which reported that the older age group is more prone to COVID-19 compare to the younger age group [3,17,18]. In recent report from hospitals in Jakarta, Indonesia, the age-specific mortalities due to COVID-19 infection were 11% for <5 years, 2.5% for 5–19 years, 7.9% for 20–60 years, and 25.5% for >60 years [14,18]. Several probabilities which affect transmission in children include having fewer outdoor activities, no face-to-face school learning, and undertaking less domestic and international travel making this age group less likely to contract the virus [19]. Since March 17, 2020, most of schools and universities in Indonesia have applied the full distance learning [20].

We found that non-healthcare workers have higher risk of COVID-19 compared to healthcare workers with the odds of COVID-19 for non-healthcare workers were almost triple the risk of healthcare workers. Positivity rates are likely affected by the nature of the individuals tested, as shown by the higher positivity rates from hospital-based specimens compared to PHC-sourced specimens. For hospitals, the majority of the samples were collected from

COVID-19 suspected individuals. On the other hand, PHC samples were mostly obtained from individuals under contact tracing in collaboration with local health agency. Moreover, the Indonesian government have started to vaccinate the population since mid-January 2021 by prioritizing frontline medical and public workers as well as older age group. More than 50 million people had received first dose, and half of them had received the second dose of COVID-19 vaccine per August 15, 2021 [15]. This warrant further studies to describe the prevalence of COVID-19 after the mass vaccination campaign. There are some limitations to our study: This study was a retrospective assessment on data of specimens submitted for routine hospital COVID-19 contact tracing and there were missing data including symptoms and patient outcome, that could alter the characteristic of COVID-19 data presentation. This study we provided data in the early of pandemic when testing capacity still limited. The data we have collected during the past year could be used to picture the overall prevalence of COVID-19 in Indonesia, and also as a baseline in evaluating the development of this pandemic in Indonesia.

## Acknowledgments

We thank multiple health facilities in different regions of Indonesia for submitting respiratory specimens.

We also thank the WASCOVE (Waspada COVID-19 Lembaga Biologi Molekuler Eijkman) team, composed by:

Lead author: Wuryantari Setiadi tari@eijkman.go.id.

Eijkman Institute for Molecular Biology, Jakarta: AA Raka Pratama, Agatha Mia Puspitasari, Ageng Wiyatno, Aghnianditya Kresno Dewantari, Ari Satyagraha, Arkasha Sadewa, Bertha Letizia Utami, Billy Witanto, Chairin Nisa Ma'roef, Chelzie Crenna Darusallam, Chrysantine Paramayuda, Clarissa Asha Febinia, Decy Subekti, Dendi Hadi Permana, Dhita Prabasari Wibowo, Eva Maria Manullang, Evira Cahya Putri, Faiza Az Zahra, Farahana Kresno Dewayanti, Fauzyah Fadlan, Firman Prathama Idris, Gladis R. Hutahaen, Hanifah Fajri Maharani Putri, Hannie Dewi Hadiani Kartapradja, Herawati Sudoyo, Hidayat Trimarsanto, I Made Artika, Indah Delima, Iskandar Alisyahbana Adnan, Jessica Rodearni Saragih, Kartika Saraswati, Korrie Salsabila, Leily Trianty, Lenny Lia Ekawati, Leonard, Leppa Shahrani, Lidwina Priliani, Lydia Visita Pangalo, Marsha S. Santoso, Miftahuddin Majid Khoeri, Muhammad Rezki Razak, Novi Dwi Susilowati, Nunung Nuraini, Rahmadania Marita Joesoef', Rifqi Risandi, Ristya Amalia, Saraswati Soebianto, Sinta Hamidatus Saidah, Tina Kusumaningrum, Ungke Antonjaya, Willy Agustine, Winahyu Handayani, Windy Joanmawanti, Wisiva Tofriska Paraimaswari, Wisnu Tafroji, Yayah Winarti, Yora Permata Dewi, Yulia Widyasanty

## Author Contributions

**Conceptualization:** Wuryantari Setiadi, Khin Saw Aye Myint, Safarina G. Malik, Amin Soebandrio.

**Data curation:** Wuryantari Setiadi, Ismail Ekoprayitno Rozi, Wa Ode Dwi Daningrat, Benediktus Yohan, Frilasita Aisyah Yudhaputri, Sukma Oktavianthi.

**Formal analysis:** Ismail Ekoprayitno Rozi, Dodi Safari, Wa Ode Dwi Daningrat, Edison Johar, Karina Dian Lestari, Sukma Oktavianthi.

**Funding acquisition:** Frilasita Aisyah Yudhaputri, Khin Saw Aye Myint, Safarina G. Malik.

**Investigation:** Wuryantari Setiadi, Ismail Ekoprayitno Rozi, Dodi Safari, Edison Johar, Frilasita Aisyah Yudhaputri, Karina Dian Lestari, Safarina G. Malik.

**Methodology:** Wuryantari Setiadi, Ismail Ekoprayitno Rozi, Dodi Safari, Wa Ode Dwi Daningrat, Edison Johar.

**Software:** Ismail Ekoprayitno Rozi, Karina Dian Lestari.

**Supervision:** Safarina G. Malik, Amin Soebandrio.

**Validation:** Ismail Ekoprayitno Rozi, Dodi Safari, Edison Johar, Benediktus Yohan, Sukma Oktavianthi, Khin Saw Aye Myint.

**Visualization:** Dodi Safari, Wa Ode Dwi Daningrat, Edison Johar, Karina Dian Lestari, Sukma Oktavianthi.

**Writing – original draft:** Wuryantari Setiadi, Dodi Safari, Wa Ode Dwi Daningrat, Edison Johar, Benediktus Yohan, Karina Dian Lestari, Khin Saw Aye Myint, Safarina G. Malik.

**Writing – review & editing:** Wuryantari Setiadi, Dodi Safari, Wa Ode Dwi Daningrat, Edison Johar, Benediktus Yohan, Karina Dian Lestari, Khin Saw Aye Myint, Safarina G. Malik, Amin Soebandrio.

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
