## [Decision Letter · Decision Letter 0]

14 Feb 2022

PONE-D-21-28166Prevalence and epidemiological characteristics of COVID-19 after one year of pandemic in Jakarta and neighbouring areas, Indonesia: A single center studyPLOS ONE

Dear Dr. Amin Soebanddrio

Thank you for submitting your manuscript to PLOS ONE. After careful consideration, we feel that it has merit but does not fully meet PLOS ONE’s publication criteria as it currently stands. Therefore, we invite you to submit a revised version of the manuscript that addresses the points raised during the review process.

There are some suggestion for revision in the text.Please follow the instruction from the reviewer. Please add the novelty and limitation in your study in the discussion section. 

Please submit your revised manuscript by 28 February 2022. If you will need more time than this to complete your revisions, please reply to this message or contact the journal office at plosone@plos.org. Please include the following items when submitting your revised manuscript:A rebuttal letter that responds to each point raised by the academic editor and reviewer(s). You should upload this letter as a separate file labeled 'Response to Reviewers'.A marked-up copy of your manuscript that highlights changes made to the original version. You should upload this as a separate file labeled 'Revised Manuscript with Track Changes'.An unmarked version of your revised paper without tracked changes. You should upload this as a separate file labeled 'Manuscript'.If applicable, we recommend that you deposit your laboratory protocols in protocols.io to enhance the reproducibility of your results. Protocols.io assigns your protocol its own identifier (DOI) so that it can be cited independently in the future. For instructions see: https://journals.plos.org/plosone/s/submission-guidelines#loc-laboratory-protocols. Additionally, PLOS ONE offers an option for publishing peer-reviewed Lab Protocol articles, which describe protocols hosted on protocols.io. Read more information on sharing protocols at https://plos.org/protocols?utm_medium=editorial-email&utm_source=authorletters&utm_campaign=protocols.

We look forward to receiving your revised manuscript.

Kind regards,

Rizaldy Taslim Pinzon

Academic Editor

PLOS ONE

" ext-link-type="uri" xlink:type="simple">https://journals.plos.org/plosone/s/fileid=ba62/PLOSOne_formatting_sample_title_authors_affiliations.pdf".

2. In the ethics statement in the manuscript and in the online submission form, please provide additional information about the patient records used in your retrospective study, including: a) whether all data were fully anonymized before you accessed them. If the ethics committee waived the need for informed consent, or patients provided informed written consent to have data from their medical records used in research, please include this information.

“This work was supported by The Ministry of Research and Technology/National Research and Innovation Agency, Republic of Indonesia. The PCR reagents to test outbreak specimens were provided by U.S. Centers for Disease Control and Prevention (US CDC), Indonesian National Board for Disaster Management (Badan Nasional Penanggulangan Bencana; BNPB), Embassy of New Zealand in Indonesia, and Indonesian States Intelligence Agency (Badan Intelijen Negara Republik Indonesia; BIN).”

“This work was supported by The Ministry of Research and Technology/National Research and Innovation Agency, Republic of Indonesia. The PCR reagents to test outbreak specimens were provided by U.S. Centers for Disease Control and Prevention (US CDC), Indonesian National Board for Disaster Management (Badan Nasional Penanggulangan Bencana; BNPB), Embassy of New Zealand in Indonesia, and Indonesian States Intelligence Agency (Badan Intelijen Negara Republik Indonesia; BIN).”

“This work was supported by The Ministry of Research and Technology/National Research and Innovation Agency, Republic of Indonesia. The PCR reagents to test outbreak specimens were provided by U.S. Centers for Disease Control and Prevention (US CDC), Indonesian National Board for Disaster Management (Badan Nasional Penanggulangan Bencana; BNPB), Embassy of New Zealand in Indonesia, and Indonesian States Intelligence Agency (Badan Intelijen Negara Republik Indonesia; BIN).”

5. One of the noted authors is a group or consortium [Wascove team]. In addition to naming the author group, please list the individual authors and affiliations within this group in the acknowledgments section of your manuscript. Please also indicate clearly a lead author for this group along with a contact email address.

7. We note that [Figure 1 ] in your submission contain [map/satellite] images which may be copyrighted. All PLOS content is published under the Creative Commons Attribution License (CC BY 4.0), which means that the manuscript, images, and Supporting Information files will be freely available online, and any third party is permitted to access, download, copy, distribute, and use these materials in any way, even commercially, with proper attribution. For these reasons, we cannot publish previously copyrighted maps or satellite images created using proprietary data, such as Google software (Google Maps, Street View, and Earth). For more information, see our copyright guidelines: http://journals.plos.org/plosone/s/licenses-and-copyright.

Reviewers' comments:

Reviewer's Responses to Questions

**Comments to the Author**

1. Is the manuscript technically sound, and do the data support the conclusions?

Reviewer #1: Yes

2. Has the statistical analysis been performed appropriately and rigorously? 

Reviewer #1: Yes

3. Have the authors made all data underlying the findings in their manuscript fully available?

Reviewer #1: Yes

4. Is the manuscript presented in an intelligible fashion and written in standard English?

Reviewer #1: Yes

5. Review Comments to the Author

Reviewer #1: Mr. Amin Soebandrio, M, PhD.

please make the correction of no of specimen. Refer the no of specimens in abstract and No. of specimens in line no 102,

please add a comma between fever and cough in line no 127

6. PLOS authors have the option to publish the peer review history of their article (what does this mean?). If published, this will include your full peer review and any attached files.

Reviewer #1: No

---

## [Author Response · Author response to Decision Letter 0]

20 Mar 2022

Rebuttal Letter

1. There are some suggestion for revision in the text.

RESPONSE: Thank you for your suggestion. We revised it

2. Please follow the instruction from the reviewer. 

RESPONSE: We revised it based on the instruction from the reviewer

3. Please add the novelty and limitation in your study in the discussion section. 

RESPONSE: Thank you for your suggestion. We add the novelty and limitation in our study in the discussion section as below (Line 191-198):

…..”There are some limitations to our study: This study was a retrospective assessment on data of specimens submitted for routine hospital COVID-19 contact tracing and there were missing data including symptoms and patient outcome, that could alter the characteristic of COVID-19 data presentation. This study we provided data in the early of pandemic when testing capacity still limited. The data we have collected during the past year could be used to picture the overall prevalence of COVID-19 in Indonesia, and also as a baseline in evaluating the development of this pandemic in Indonesia”…..

4. In the ethics statement in the manuscript and in the online submission form, please provide additional information about the patient records used in your retrospective study, including: a) whether all data were fully anonymized before you accessed them. If the ethics committee waived the need for informed consent, or patients provided informed written consent to have data from their medical records used in research, please include this information.

RESPONSE: We add the ethics statement and the patient records used in the manuscript as below (Line 74-76):

…..” This study was performed in accordance with the human subject protection guidance provided by the Eijkman Institute Research Ethics Committee No. 127. All data were fully anonymized before accessing the database.”…..

“This work was supported by The Ministry of Research and Technology/National Research and Innovation Agency, Republic of Indonesia. The PCR reagents to test outbreak specimens were provided by U.S. Centers for Disease Control and Prevention (US CDC), Indonesian National Board for Disaster Management (Badan Nasional Penanggulangan Bencana; BNPB), Embassy of New Zealand in Indonesia, and Indonesian States Intelligence Agency (Badan Intelijen Negara Republik Indonesia; BIN).” We note that you have provided funding information that is not currently declared in your Funding Statement. However, funding information should not appear in the Acknowledgments section or other areas of your manuscript. We will only publish funding information present in the Funding Statement section of the online submission form. Please remove any funding-related text from the manuscript and let us know how you would like to update your Funding Statement. Currently, your Funding Statement reads as follows:

“This work was supported by The Ministry of Research and Technology/National Research and Innovation Agency, Republic of Indonesia. The PCR reagents to test outbreak specimens were provided by U.S. Centers for Disease Control and Prevention (US CDC), Indonesian National Board for Disaster Management (Badan Nasional Penanggulangan Bencana; BNPB), Embassy of New Zealand in Indonesia, and Indonesian States Intelligence Agency (Badan Intelijen Negara Republik Indonesia; BIN).” Please include your amended statements within your cover letter; we will change the online submission form on your behalf.

RESPONSE: Thank you for your suggestion. We add our amended statements within our cover letter.

Financial Disclosure:

This work was supported by The Ministry of Research and Technology/National Research and Innovation Agency, Republic of Indonesia. The PCR reagents to test outbreak specimens were provided by U.S. Centers for Disease Control and Prevention (US CDC), Indonesian National Board for Disaster Management (Badan Nasional Penanggulangan Bencana; BNPB), Embassy of New Zealand in Indonesia, and Indonesian States Intelligence Agency (Badan Intelijen Negara Republik Indonesia; BIN). The funders had no role in study design, data collection and analysis, decision to publish, or preparation of the manuscript

6. Thank you for stating the following financial disclosure:

“This work was supported by The Ministry of Research and Technology/National Research and Innovation Agency, Republic of Indonesia. The PCR reagents to test outbreak specimens were provided by U.S. Centers for Disease Control and Prevention (US CDC), Indonesian National Board for Disaster Management (Badan Nasional Penanggulangan Bencana; BNPB), Embassy of New Zealand in Indonesia, and Indonesian States Intelligence Agency (Badan Intelijen Negara Republik Indonesia; BIN).”

Please state what role the funders took in the study. If the funders had no role, please state: ""The funders had no role in study design, data collection and analysis, decision to publish, or preparation of the manuscript."" If this statement is not correct you must amend it as needed. Please include this amended Role of Funder statement in your cover letter; we will change the online submission form on your behalf.

7. RESPONSE: Thank you for your suggestion. We add our amended statements within our cover letter.

Financial Disclosure:

This work was supported by The Ministry of Research and Technology/National Research and Innovation Agency, Republic of Indonesia. The PCR reagents to test outbreak specimens were provided by U.S. Centers for Disease Control and Prevention (US CDC), Indonesian National Board for Disaster Management (Badan Nasional Penanggulangan Bencana; BNPB), Embassy of New Zealand in Indonesia, and Indonesian States Intelligence Agency (Badan Intelijen Negara Republik Indonesia; BIN). The funders had no role in study design, data collection and analysis, decision to publish, or preparation of the manuscript

One of the noted authors is a group or consortium [Wascove team]. In addition to naming the author group, please list the individual authors and affiliations within this group in the acknowledgments section of your manuscript. Please also indicate clearly a lead author for this group along with a contact email address.

RESPONSE: Membership of the Wascove team is provided in the Acknowledgments.

“We also thank the WASCOVE (Waspada COVID-19 Lembaga Biologi Molekuler Eijkman) team, composed by: 

Lead author: Wuryantari Setiadi tari@eijkman.go.id

Eijkman Institute for Molecular Biology, Jakarta: AA Raka Pratama, Agatha Mia Puspitasari, Ageng Wiyatno, Aghnianditya Kresno Dewantari, Ari Satyagraha, Arkasha Sadewa, Bertha Letizia Utami, Billy Witanto, Chairin Nisa Ma'roef, Chelzie Crenna Darusallam, Chrysantine Paramayuda, Clarissa Asha Febinia, Decy Subekti, Dendi Hadi Permana, Dhita Prabasari Wibowo, Eva Maria Manullang, Evira Cahya Putri, Faiza Az Zahra, Farahana Kresno Dewayanti, Fauzyah Fadlan, Firman Prathama Idris, Gladis R. Hutahaen, Hanifah Fajri Maharani Putri, Hannie Dewi Hadiani Kartapradja, Herawati Sudoyo, Hidayat Trimarsanto, I Made Artika, Indah Delima, Iskandar Alisyahbana Adnan, Jessica Rodearni Saragih, Kartika Saraswati, Korrie Salsabila, Leily Trianty, Lenny Lia Ekawati, Leonard, Leppa Shahrani, Lidwina Priliani, Lydia Visita Pangalo, Marsha S. Santoso, Miftahuddin Majid Khoeri, Muhammad Rezki Razak, Novi Dwi Susilowati, Nunung Nuraini, Rahmadania Marita Joesoef", Rifqi Risandi, Ristya Amalia, Saraswati Soebianto, Sinta Hamidatus Saidah, Tina Kusumaningrum, Ungke Antonjaya, Willy Agustine, Winahyu Handayani, Windy Joanmawanti, Wisiva Tofriska Paraimaswari, Wisnu Tafroji, Yayah Winarti, Yora Permata Dewi, Yulia Widyasanty”

8. Please include your full ethics statement in the ‘Methods’ section of your manuscript file. In your statement, please include the full name of the IRB or ethics committee who approved or waived your study, as well as whether or not you obtained informed written or verbal consent. If consent was waived for your study, please include this information in your statement as well.

RESPONSE: We add the ethics statement and the patient records used in the manuscript as below (Line 74-76):

…..” This study was performed in accordance with the human subject protection guidance provided by the Eijkman Institute Research Ethics Committee No. 127. All data were fully anonymized before accessing the database.”…..

9. We note that [Figure 1 ] in your submission contain [map/satellite] images which may be copyrighted. All PLOS content is published under the Creative Commons Attribution License (CC BY 4.0), which means that the manuscript, images, and Supporting Information files will be freely available online, and any third party is permitted to access, download, copy, distribute, and use these materials in any way, even commercially, with proper attribution. For these reasons, we cannot publish previously copyrighted maps or satellite images created using proprietary data, such as Google software (Google Maps, Street View, and Earth). For more information, see our copyright guidelines: http://journals.plos.org/plosone/s/licenses-and-copyright.

We require you to either (1) present written permission from the copyright holder to publish these figures specifically under the CC BY 4.0 license, or (2) remove the figures from your submission: You may seek permission from the original copyright holder of Figure 1 to publish the content specifically under the CC BY 4.0 license. We recommend that you contact the original copyright holder with the Content Permission Form (http://journals.plos.org/plosone/s/file?id=7c09/content-permission-form.pdf) and the following text: “I request permission for the open-access journal PLOS ONE to publish XXX under the Creative Commons Attribution License (CCAL) CC BY 4.0 (http://creativecommons.org/licenses/by/4.0/). Please be aware that this license allows unrestricted use and distribution, even commercially, by third parties. Please reply and provide explicit written permission to publish XXX under a CC BY license and complete the attached form.” Please upload the completed Content Permission Form or other proof of granted permissions as an ""Other"" file with your submission. In the figure caption of the copyrighted figure, please include the following text: “Reprinted from [ref] under a CC BY license, with permission from [name of publisher], original copyright [original copyright year].”

If you are unable to obtain permission from the original copyright holder to publish these figures under the CC BY 4.0 license or if the copyright holder’s requirements are incompatible with the CC BY 4.0 license, please either i) remove the figure or ii) supply a replacement figure that complies with the CC BY 4.0 license. Please check copyright information on all replacement figures and update the figure caption with source information. If applicable, please specify in the figure caption text when a figure is similar but not identical to the original image and is therefore for illustrative purposes only.The following resources for replacing copyrighted map figures may be helpful:

RESPONSE: We revised the Figure 1 to meet to meet PLOS ONE's guidelines using http://www.naturalearthdata.com/

RESPONSE: We reviewed our reference list. We confirmed that is complete and correct

11. Reviewer #1: Mr. Amin Soebandrio, M, PhD.

please make the correction of no of specimen. Refer the no of specimens in abstract and No. of specimens in line no 102,

RESPONSE: We revised it

12. please add a comma between fever and cough in line no 127

RESPONSE: We revised it

---

## [Editor Report · Decision Letter 1]

4 Apr 2022

PONE-D-21-28166R1Prevalence and epidemiological characteristics of COVID-19 after one year of pandemic in Jakarta and neighbouring areas, Indonesia: A single center studyPLOS ONE

Dear Dr.  Amin Soebandrio 

Thank you for submitting your manuscript to PLOS ONE. After careful consideration, we feel that it has merit but does not fully meet PLOS ONE’s publication criteria as it currently stands. Therefore, we invite you to submit a revised version of the manuscript that addresses the points raised during the review process. Some minor revision and clarification: The data in the text mention that : The province that contribute for more than 60% cases are Jakarta (21.3%), West Java (16.8%), Central Java (11.6%), East Java (9.4%), and East Kalimantan (3.7%). The figure 1 do not showed any data from East Kalimantan. Please add not only symptoms, but also radiological finding for pneumonia in the text and figure 3. 

We look forward to receiving your revised manuscript.

Kind regards,

Rizaldy Taslim Pinzon

Academic Editor

PLOS ONE
---

## [Author Response · Author response to Decision Letter 1]

11 Apr 2022

PONE-D-21-28166R1

Prevalence and epidemiological characteristics of COVID-19 after one year of pandemic in Jakarta and neighbouring areas, Indonesia: A single center studyPLOS ONEDear 

1. Some minor revision and clarification: The data in the text mention that: The province that contribute for more than 60% cases are Jakarta (21.3%), West Java (16.8%), Central Java (11.6%), East Java (9.4%), and East Kalimantan (3.7%). The figure 1 do not showed any data from East Kalimantan.

RESPONSE: Thank you for your comments. These data were cited from the National Committee for Corona Virus Diseases 2019 Handling and Economic Recovery reported on August 2021 (In discussion part: Line 155-161). Meanwhile, Figure 1 showed data for current study (Line 107-111). We cited the report from the National Committee to provide comparison (Reference No. 15). We have revised the text and added citation and reference accordingly.

2. Please add not only symptoms, but also radiological finding for pneumonia in the text and figure 3.

RESPONSE: Thank you for your suggestion. We have revised the text to include pneumonia based on chest x-ray results and figure 3 has been updated accordingly. Please see line 130-136 and Figure 3:

……..”Chest X-ray data were available from 845 out of 10130 positive cases, which were reported as: pneumonia 679/845 (80.4%), other abnormalities (including bronchitis, pleural effusion, lung abscess, and increased bronchovascular markings) 46/845 (5.4%), and normal 120/845 (14.2%). Most of SARS-CoV-2 cases were asymptomatic (74.0%). Among symptomatic cases of SARS-CoV-2, we found that the most common reported clinical symptoms (or combination of symptoms) (n=1,577) were cough (20.9%), a combination of both cough fever (7.0%), and fever (6.8%) (Figure 3).”……..

---

## [Editor Report · Decision Letter 2]

26 Apr 2022

Prevalence and epidemiological characteristics of COVID-19 after one year of pandemic in Jakarta and neighbouring areas, Indonesia: A single center study

PONE-D-21-28166R2

Dear Dr. Amin Soebandrio

We’re pleased to inform you that your manuscript has been judged scientifically suitable for publication and will be formally accepted for publication once it meets all outstanding technical requirements.

Kind regards,

Rizaldy Taslim Pinzon

Academic Editor

PLOS ONE
---

## [Editor Report · Acceptance letter]

3 May 2022

PONE-D-21-28166R2 

Prevalence and epidemiological characteristics of COVID-19 after one year of pandemic in Jakarta and neighbouring areas, Indonesia: A single center study 

Dear Dr. Soebandrio:

I'm pleased to inform you that your manuscript has been deemed suitable for publication in PLOS ONE. Congratulations! Your manuscript is now with our production department. 

Kind regards, 

on behalf of

Dr. Rizaldy Taslim Pinzon 

Academic Editor

PLOS ONE